# Differential Early Performance of Two Underplanted Hardwood Tree Species Following Restoration Treatments in High-Graded Temperate Rainforests

**Daniel P. Soto [1,*]** , **Pablo J. Donoso [2]** , **Angélica Vásquez-Grandón [2]** ,
**Mauricio González-Chang [1]** **and Christian Salas-Eljatib [3,4]**

[1] Departamento de Ciencias Naturales y Tecnología, Universidad de Aysén, Coyhaique 5950000, Chile; mauricio.gonzalez@uaysen.cl

[2] Instituto de Bosques y Sociedad, Universidad Austral de Chile, Valdivia 5090000, Chile; pdonoso@uach.cl (P.J.D.); angevasquez.ingfor@gmail.com (A.V.-G.)

[3] Centro de Modelación y Monitoreo de Ecosistemas, Universidad Mayor, Santiago 8320000, Chile; cseljatib@gmail.com

[4] Laboratorio de Biometría, Universidad de La Frontera, Temuco 4780000, Chile

[*] Correspondence: daniel.soto@uaysen.cl; Tel.: +56-9-79983561

**Abstract:** Raulí (*Nothofagus alpina* (Poepp. & Endl.)) and Ulmo (*Eucryphia cordifolia* Cav.) are mid-tolerant tree species in the Coihue-Raulí-Tepa (ca. 0.55 mill ha) and Evergreen (ca. 4.1 mill ha) forest types in south-central Chile, respectively. These species have been selectively logged in old-growth forests especially during the 20th century, Raulí mostly for its highly valuable timber, and Ulmo for its highly demanded firewood and bark for the tannery industry. Natural regeneration of these species occurs mostly through canopy gaps, but it can be retarded, or even inhibited, when the cover of the understory vegetation becomes unusually dense, such as in high-graded forests. Although underplanting is possible for these species, the knowledge about their growth in forest understories is scarce, and necessary to inform restoration programs. Therefore, we evaluated short-term responses (two years) of underplanted containerized seedlings in root-collar diameter, height, stem volume, and in the slenderness index, as a function of canopy openness (%, continuous variable) and three restoration treatments (categorical variables, plus one control treatment) at two different sites with high-graded old-growth forests for each forest type. By using generalized linear mixed-effects models (GLMMs) we determined that Raulí was more sensitive to the influence of both canopy openness and restoration treatments, while Ulmo was mostly influenced by canopy openness. Specifically, Raulí was positively influenced by canopy openness and restoration treatments in all response variables except for the slenderness index. Conversely, Ulmo was influenced by canopy openness in all response variables except the slenderness index, which was influenced by both predictor variables (canopy openness and restoration treatments). Thus, prospects for restoration with these species are discussed, including possible ontogenetic changes in their responses to light that may demand continuous silvicultural operations to recover the productive and functional roles of these species in these forest ecosystems.

**Keywords:** arrested succession; high-graded forests; *Eucryphia cordifolia*; forest restoration; *Nothofagus alpina*; underplanted

## 1. Introduction

Forest degradation is estimated to affect nearly one-half of the world's forests, which are less biodiverse and productive than well-conserved forests [1,2]. Commonly, forest degradation delays

or arrests forest recovery in the mid- to the long-term and reduces some ecological functions and processes throughout successional development [1,3]. Arrested succession can occur when, for example, some undesirable and pervasive understory vegetation is released and then occupies the understory layer over long periods, retarding forest recovery [3–5]. Heavily degraded forests may lack the opportunity to regenerate many species, especially if there are few seed sources and the understory has attained a great cover of shrubs that compete for light and water required by these species during the regeneration process [3,5]. Underplanting may become the main option to restore these species in these human-disturbed forest ecosystems [6–10]. In these cases, without appropriate silviculture protocols, the risks of potential undesirable outcomes (e.g., ecosystem disservices [11,12]) and undesired successional pathways (e.g., arrested succession) may be increased [3,6,8]. Silvicultural practices can reverse these potential scenarios through the control of undesirable understory vegetation and underplanting when seed sources and natural tree regeneration are lacking [8–10,13].

Two large forest types in Chile, the Evergreen (ca. 4.1 mill ha) and the Coihue-Raulí-Tepa (ca. 0.55 mill ha) [14] have been severely high-graded since the last two centuries due to the high timber value of tree species within these forest types [15–17], especially Ulmo (*Eucryphia cordifolia*) in the evergreen forest type, and Raulí (*Nothofagus alpina*) in the Coihue-Raulí-Tepa forest type. These species have been severely exploited through selective harvesting [15,16,18], mainly from old-growth forests during the 20th century, Raulí mostly for its highly valuable timber, and Ulmo mostly for its highly demanded firewood and bark for the tannery industry, in addition to its timber value [16]. These tree species are mid-tolerant to shade and eutrophic in terms of soil requirements [19]. Since they can attain old ages and large sizes, they play key ecosystem roles, having a major influence upon ecological functions and processes, and in the distribution and abundance of tree populations and communities [20]. Both species play a pivotal role in community reorganization after disturbance and through successional development to their forest types [21–23].

Studies about the regeneration dynamics for these species illustrate that they generally follow the gap-phase regeneration mode [24,25], when gaps are large enough to allow their establishment, which is typically around 1000 m$^2$ [23,26]. Therefore, both species are able to regenerate after partial overstory disturbances, especially if understory competition is moderate [27,28]. However, the fern *Lophosoria quadripinnata* (J. F. Gmelin) C. Chr. in the Evergreen forest type [29], and bamboo species (*Chusquea* spp.) in both forest types [3,30,31], among other species, may become highly competitive in high-graded old-growth forest, and may delay forest recovery over long periods of time [3].

The forest where Rauli and Ulmo occur have been modified by logging or replaced by forest plantations of exotic species or even conventional agricultural systems, and their recovery is essential to approach a more sustainable forest sector in south-central Chile [32–34]. In an effort to restore these landscapes, plantation or restoration with native species play a fundamental role. Therefore, research regarding the development of species like Raulí and Ulmo when underplanted in high-graded native forests or in other forest restoration efforts is essential [17]. However, this knowledge is still scarce [9]. Furthermore, information about the performance of Raulí is quite limited to local scales [9,10,17,35], and for Ulmo, there are few evaluations of its performance under partial shade conditions [36,37]. Field experiments are the preferred approach to inform environmental and forestry policies to aid practitioners and landowners to understand the potential of this restoration tool [17,34,38]. In this study, we assessed the early-growth of underplanted Ulmo and Raulí seedlings as a function of canopy openness and initial restoration treatments during the first two growing seasons in high-graded old-growth forests in the Evergreen and the Coigue-Raulí-Tepa forest types. We hypothesized that canopy openness and the restoration treatments differentially enhance the early growth of Ulmo and Raulí in these forest types.

## 2. Materials and Methods

### 2.1. Study Sites

The present study was carried out in two of the major forest types within the Valdivian Temperate Rainforests of South America in south-central Chile, the Evergreen and the Coihue-Raulí-Tepa [39] (Figure 1). We selected two sites for each forest type in which we identified old-growth stands that were high-graded 30–50 years ago (PJ Donoso, personal observation). These stands have a scattered distribution of medium- to large-sized trees, and a patchy distribution of poled-sized trees mostly of poor quality of low-value tree species and dense thickets of *Chusquea* spp. (*C. quila* in the Evergreen forests and *C. coleou* in the Coihue-Raulí-Tepa forests), and ferns or other understory competing plant species (especially in Evergreen forest type). Different competing understory species and poor tree regeneration are also noted [40]. The general characteristics of the studied forest types are presented in Table 1.

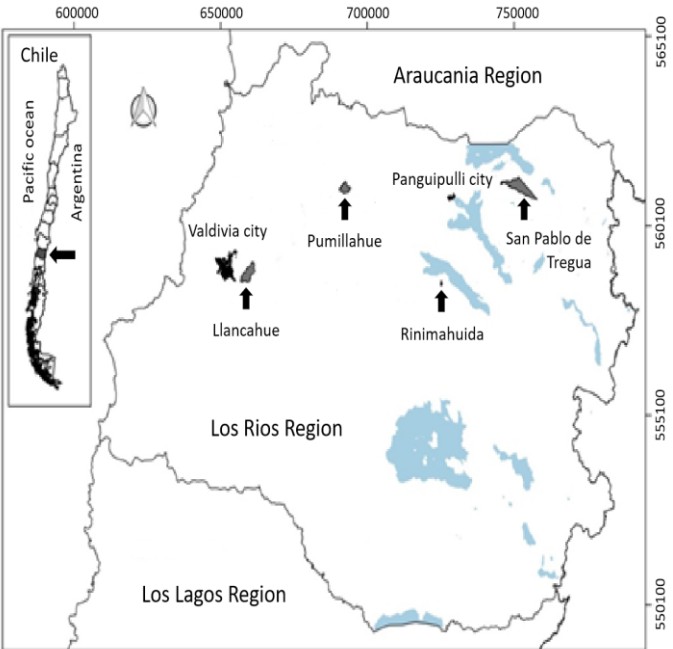

**Figure 1.** Study sites location in the landscape of south-central Chile: Llancahue, Pumilahue, Rinimahuida, and San Pablo de Tregua. Blue areas are major lakes.

The sites for the evergreen forest type were located in two public forest reserves, Llancahue and Pumillahue, both in the Loncoche's transversal mountain range in the Chilean intermediate depression. Similar climate (coastal oceanic with a Mediterranean influence according to Koeppen classification, [40]) but some soil differences appeared between both sites (Table 1). In Llancahue, the soil series corresponds to Los Ulmos (Typic Paleudults: Ultisol), which are dominated by clay. At Pumillahue, the soil series is Correltúe (Andic Palehumults: Ultisol) characterized by silt loam soils [41]. These soils are derived from old volcanic ashes and have suffered intense weathering processes that lead to low soil fertility (Table 1). Correltúe has a better rooting development compared to Los Ulmos, and also a higher proportion of organic carbon (12% and 6.5%, respectively). Correltúe also has a higher phosphorus content up to 13 ppm [41]. Therefore, the Pumillahue site has better soil properties for seeding establishment than Llancahue.

The sites for the Coihue-Raulí-Tepa forest type are located in the San Pablo de Tregua experimental forest station belonging to Universidad Austral de Chile, and in Riñimahuida (private property). Both sites are in the Andes range and correspond to the same Liquiñe soil series (Acrudoxic Hapludands: Andisol). The soil in San Pablo de Tregua has an infertile pumice horizon over basaltic-andesitic

rocks [41], presenting high phosphorus retention and aluminum levels due to the presence of alophan. In Riñimahuida, the soil has a fertile horizon of about 30 to 40 cm in depth, but its main limitation is excessive drainage, which makes it susceptible to erosion by water or wind [40,41]. As both soils have Andic properties, it is not unusual to find low bulk density, acidic soil conditions, high porosity and fast drainage [42].

**Table 1.** General characteristics of the experiment and sites.

| Location | San Pablo de Tregua | Riñimahuida | Llancahue | Pumillahue |
|---|---|---|---|---|
| **Sites Characteristics** | | | | |
| County | Panguipulli | Riñihue | Valdivia | Mariquina |
| Latitude | 39°36′21″ S | 39°50′49″ S | 39°50′56″ S | 39°38′20″ S |
| Longitude | 72°06′08″ W | 72°22′11″ W | 73°07′50″ W | 72°45′01″ W |
| Altitude (m a.s.l.) | 750 | 600 | 325 | 260 |
| **Climate Characteristics (average)** | | | | |
| Annual precipitation (mm) | 3863 | 2279 | 2100 | 2500 |
| Mean annual temperature (°C) | 11 | 13 | 12 | 12 |
| **Forest Characteristics (Mean ± Standard Deviation)** | | | | |
| Density (trees ha$^{-1}$) | 914 ± 468 | 843 ± 191 | 1135 ± 306 | 775 ± 290 |
| Basal area (m$^2$ ha$^{-1}$) | 65.5 ± 10 | 25.4 ± 4 | 48.8 ± 10 | 64.7 ± 23 |
| Dg (cm) | 31.4 ± 12 | 18.8 ± 1 | 23.4 ± 13 | 32.6 ± 5 |
| **Soil Characteristics** | | | | |
| Soil series, type and family | Liquiñe Recent volcanic ashes (Andisol) | Liquiñe Recent volcanic ashes (Andisol) | Los Ulmos Old volcanic ashes (Ultisol) | Correltúe Old volcanic ashes (Ultisol) |
| Soil thickness | <100 | <100 | <160 | <130 |
| Soil texture | Fine sandy-loam | Fine sandy-loam | clay-loam | silt-loam |
| pH (H$_2$O) | 5.5–5.7 | 5.5–5.7 | 5.0–5.4 | 4.9–5.6 |
| Bulk density (range, g/cm3) | 0.51–0.79 | 0.51–0.79 | 0.83–1.11 | 0.67–0.84 |
| Soil water availability (range, %) | 10.6–24.7 | 10.6–24.7 | 14.0–18.8 | 13.0–22.0 |
| P retention (range, %) | 94–98 | 94–98 | 91–95 | 89–97 |

Sources: [40,41,43–45]

These forests had between 775 to 914 trees per hectare and basal areas between 25.4 to 65.5 m$^2$ per hectare (Table 1). These basal areas represent between one-fourth to two-thirds of the common basal areas expected in old-growth forests in the region [43].

## 2.2. Study Design and Restoration Treatments

Four 2000 m$^2$ plots, divided in four quadrants, were established at each site. In each quadrant, three restoration treatments were randomly assigned and implemented, leaving one quadrant as a control (no treatment) (Table 2). This design was not completely orthogonal, but it represents increasing intensity to promote theoretically better conditions for the regeneration of light-demanding tree species. After implementation of the restoration treatments, a plantation was systematically conducted in each treated quadrant. In order to capture the greater variability in canopy openness, 20 seedlings per quadrant were established.

**Table 2.** Description of the restoration treatments implemented in the experiment.

| Treatments | Description and Activities |
|---|---|
| 1. Control (untreated plots) | Underplanting without restoration treatments. |
| 2. Improvement cut | Underplanting plus improvement cut of trees with lowest quality and thinning among clumped groups of low-diameter trees with the goal to homogenize light penetration into the understory to stimulate growth of seedlings. |
| 3. Improvement cut and understory vegetation control | Same as treatment 2, but including understory control in the entire plot, which included manually cutting the shrubs and piling them outside of the plots. This treatment aimed to avoid competition of seedlings with understory vegetation. |
| 4. Improvement cut, understory vegetation control and soil scarification | Same as above (3), plus manual topsoil scarification with the goal to remove the litter layer, including small woody debris (<20 cm in diameter), which was piled out of the regeneration plots. |

*2.3. Plant Material*

Seedlings used in this study had a range from 35 to 45 cm in total height (*h*) and 4 to 5 mm in root-collar diameter (*d*). Neither showed significant differences in these variables nor their stem volume index (*v*) at the establishment. Seedlings were produced in 16 m tall containers with 130 cm$^3$ in rooting volume, with a substrate of composted *Pinus radiata* (Monterey pine) bark mixed with a slow-release fertilizer (e.g., 5 kg per 1 m$^3$ of composted bark). Details about protocols for seedling production and the characteristics of these seedlings can be found in Bustos et al. [46] and Donoso et al. [47].

*2.4. Measurements*

At the time of planting and following two growing seasons, we measured root collar diameter (*d*) and height (*h*) during the dormant season (winter). Stem volume index (*v*, cm$^3$ seedling$^{-1}$) was calculated using the cone formula as in Rose and Ketchum [48]. Furthermore, we computed the slenderness index (*Slen*), at year 2, which is the ratio of *h/d*, where lower values reflect plants with better biomass distribution, therefore having a greater likelihood of better field performance [47]. Finally, we computed the periodic annual increment (*pai*) for diameter, height, and volume for each seedling, represented as $pai_d$, $pai_h$, and $pai_v$, respectively.

We computed canopy openness (CO, %) at the apex of each seedling by taking hemispherical photographs processed by the software gap light analyzer (GLA) [49]. Further details about the settings used in GLA are given by earlier studies of Donoso et al. [9,45] and Soto et al. [10].

*2.5. Statistical Analyses*

We analyzed the effects of canopy openness and the four restoration treatments on the response variables $pai_d$, $pai_h$, $pai_v$, and *Slen*. We fit mixed-effects models for each response variable, by using canopy openness and the treatments as predictor variables; meanwhile, site and the plot were considered as random effects. The mixed-effects model framework allowed us to take into account the hierarchical structure of the data, producing more efficient estimates of the variance components [50–53]. All the mixed-effects models were fitted by restricted estimated maximum likelihood. The best random variable structure to be used in this experiment was the plot nested within the site (site/plot). The results for the selection of this random structure is presented in Appendix A.

For each response variable, we compared the following four models by modifying the predictor variables: (1) canopy openness only, (2) restoration treatments only, (3) canopy openness and restoration treatments, and (4) the interaction between canopy openness and the restoration treatments. The Akaike information criteria (AIC) as well as the Bayesian information criteria (BIC) were used to compare each of the four models [54]. All statistical analyses were performed using the R packages "nlme", "lme4" [50], and "effects" [55] using R [56].

## 3. Results

Two growing seasons following the experiment establishment, both species showed a quite similar growth patterns but different responses to canopy openness and the restoration treatments (Figure 2). For example, Raulí was significantly influenced by canopy openness and the restoration treatments for root-collar diameter ($pai_d$), total height ($pai_h$), and stem volume index ($pai_v$), but not for the slenderness index (*Slen*) (Table 3, Figure 2). Conversely, Ulmo was only significantly influenced by canopy openness in $pai_d$, $pai_h$, and $pai_v$, while *Slen* was significantly influenced by both fixed effects (Table 3). Considering only the canopy openness at a level of 40% (dashed lines in Figure 2), similar growth trends were observed for both species, but Ulmo had better growth and narrower error bands (95% confidence intervals) in $pai_h$ and $pai_v$ than Raulí (Figure 2). The latter grew slightly better in $pai_d$ than Ulmo (Figure 2).

**Table 3.** Akaike's (AIC) and Bayesian information criterion indices (BIC) of the fitted mixed-effects models by the response variables and species. The best-supported model for each variable is presented in bold numbers.

| Response variable | Raulí Model | AIC | BIC | Ulmo AIC | BIC |
|---|---|---|---|---|---|
| $pai_d$ | 1 | −683.6 | −664.0 | **−1297.7** | **−1277.1** |
| $pai_d$ | 2 | −637.0 | −609.6 | −1235.2 | −1206.3 |
| $pai_d$ | 3 | **−694.9** | **−663.6** | −1294.2 | −1261.2 |
| $pai_d$ | 4 | −690.0 | −646.9 | −1292.6 | −1247.3 |
| $pai_h$ | 1 | 3162.5 | 3182.1 | **3544.3** | **3564.9** |
| $pai_h$ | 2 | 3152.9 | 3180.3 | 3567.5 | 3596.3 |
| $pai_h$ | 3 | **3124.9** | **3156.3** | 3548.3 | 3581.2 |
| $pai_h$ | 4 | 3127.2 | 3170.3 | 3635.4 | 3680.7 |
| $pai_v$ | 1 | 2225.4 | 2244.9 | **2225.2** | **2258.2** |
| $pai_v$ | 2 | 2246.1 | 2273.5 | 2306.3 | 2335.2 |
| $pai_v$ | 3 | **2176.7** | **2208.0** | 2228.4 | 2248.0 |
| $pai_v$ | 4 | 2180.5 | 2223.6 | 2230.0 | 2275.3 |
| *Slen* | 1 | **3440.1** | **3459.7** | 4039.5 | 4060.1 |
| *Slen* | 2 | 3464.9 | 3492.3 | 4047.4 | 4076.2 |
| *Slen* | 3 | 3443.2 | 3474.6 | **4035.3** | **4068.2** |
| *Slen* | 4 | 3443.6 | 3486.7 | 4038.3 | 4083.7 |

Note: $pai_d$: periodic annual increment in root-collar diameter; $pai_h$: periodic annual increment in total height; $pai_v$: periodic annual increment in stem volume; *Slen*: slenderness index.

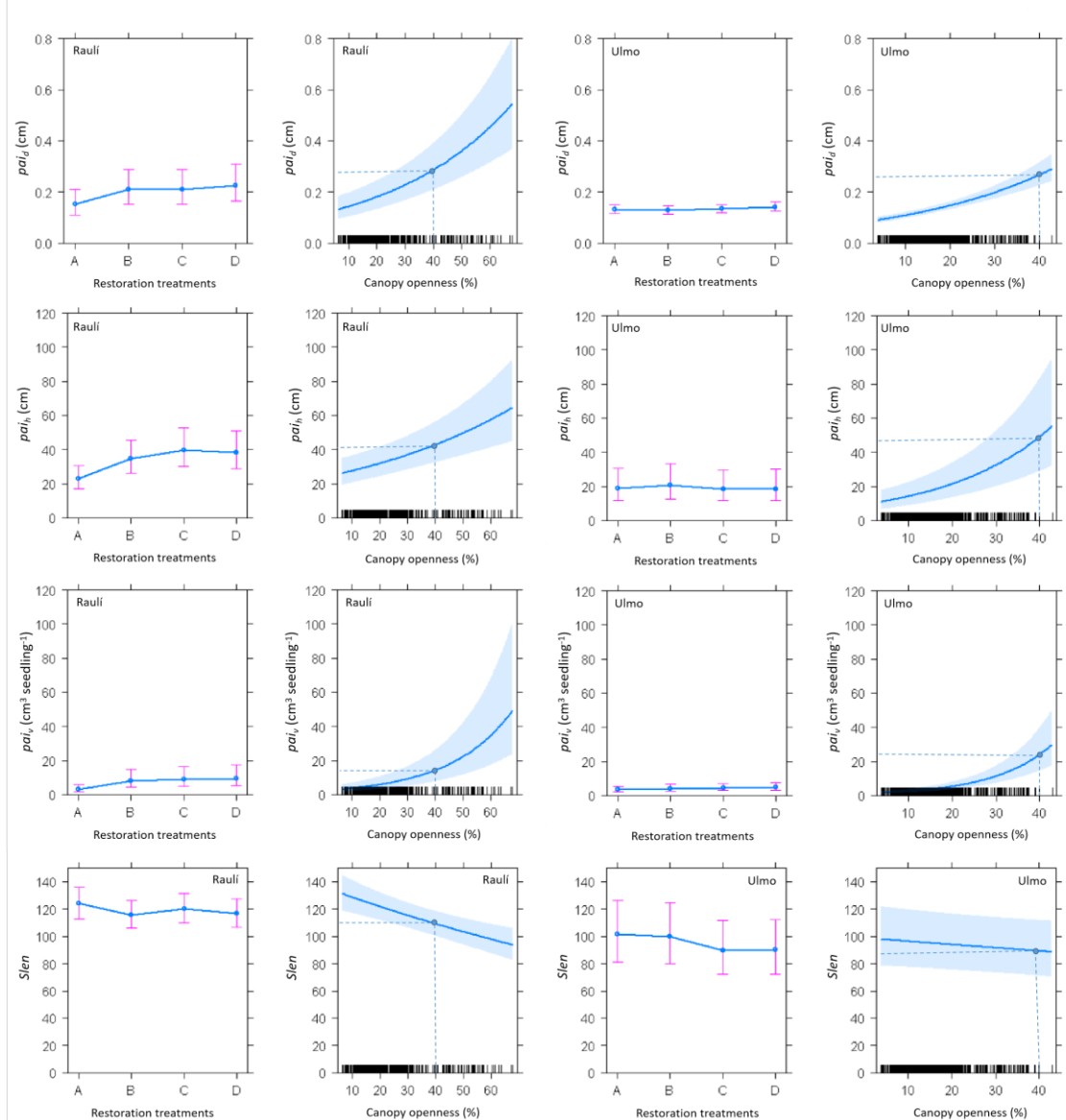

**Figure 2.** Expected value for each response variable for each species as a function of canopy openness, based on our proposed mixed-effects models. A 95% confidence interval is shown for each model by shadowed areas around the expected value. Dashed lines show the canopy openness at a level of 40%. The plots related to the restoration treatments show the mean and standard deviation for each of the four response variables measured for each species. The plots related to canopy openness show the fitted generalized linear mixed model (GLMM) with shadowed areas as the confidence intervals (95%). The plots related to restoration treatments show A: control (untreated plots), B: Improvement cut, C: Improvement cut and understory vegetation control, D: Improvement cut, understory vegetation control, and soil scarification.

For Raulí, the restoration treatments had significant effects on $pai_d$, $pai_h$, and $pai_v$, but not on *Slen*. Specifically, the three restoration treatments evaluated here had significant effects with respect to the untreated plots (Table 3). The improvement cut (B) and improvement cut with understory vegetation control (C) treatments had 23% higher $pai_d$ than the control treatment (A). On the other hand, there was a 35.3% increase in $pai_d$ in the improvement cut, understory control, and soil scarification treatment (D) over the control (Figure 2). A similar trend was obtained for $pai_h$, where treatment C was 62.5%, treatment D was 54.1%, and treatment B was 45.8%, higher than the control (Figure 2). Treatment

outperformance compared to the control was greatest for $pai_v$, which was 150% higher in treatments C and D, and 135% higher in treatment B with respect to the control treatment (Figure 2).

For Ulmo, the restoration treatments did not have a significant effect upon $pai_d$, $pai_h$, and $pai_v$, but they did upon *Slen* (Table 3). Specifically, the control had higher *Slen* compared to the other treatments, being 2% higher than B, and 11.8% higher than treatments C and D (Figure 2).

## 4. Discussion

### 4.1. Growth Responses to Partial Light Conditions

Overall, Ulmo had a slightly better growth performance than Raulí. The periodic annual increment in $d$ ($pai_d$) was similar for both species, averaging 0.3 cm $yr^{-1}$ at 40% canopy openness, with both species showing a positive $pai_d$ response to increasing canopy openness, an expected response for these mid-tolerant species, which have similar light compensation points (Ulmo 5.3 ± 1.2 lmol $m^{-2}$ $s^{-1}$ Ulmo [57], and Raulí 7.0 ± 0.4 lmol $m^{-2}$ $s^{-1}$ [58], compared with shade-intolerant *Nothofagus* species that have values close to 20 lmol $m^{-2}$ $s^{-1}$ [58]). However, Raulí had higher variability in $pai_d$ than Ulmo, suggesting that it is more sensitive to local variation in environmental conditions, such as soil quality and canopy openness [28]. Actually, Lusk et al. [59] showed that juveniles (50–200 cm in height) of *E. cordifolia* developed mostly within a range of 20% to 70% canopy openness in old-growth forests, while Donoso [19] reported that natural regeneration of *N. alpina* has negligible growth rates under closed canopies in old-growth forests. Nevertheless, in general, both species follow the gap-regeneration mode described by Veblen et al. [25], and may reach the forest canopies in old-growth forests only if partial light conditions are available in the under- and mid-stories [19,22,23,27], such as Raulí does in partially harvested *Nothofagus*-dominated old-growth forests in the Andes [28]. Therefore, underplanting of these species would succeed within a range of partial canopy openness, and promoting seedling growth into larger size classes as they recruit into the overstory should also be accompanied by controls in canopy openness [9].

### 4.2. Mixed Silvicultural Treatments to Favor the Development of Underplanted Seedlings in High-Graded Forest

The restoration treatments conducted here had marked effects on the growth (*pai* in *d, h,* and *v*) of Raulí but not of Ulmo. Specifically, growth in height and volume for Raulí was higher in the treatment that included improvement cut and understory control, while the lowest growth rates occurred under control plots. Different studies have indirectly shown the sensitivity of Raulí and Ulmo to competition with understory vegetation [9,19,28]. Thus, Raulí is a highly demanding species in terms of niche conditions for its establishment and growing phases [3]. Similarly, Uteau and Donoso [37] suggested that Ulmo has strong competition for light in young underplanted seedlings (asymmetric competition), but that during the sapling stage, competition with its conspecific species occurs mostly for soil nutrients and water (symmetric competition). However, we did not observe this pattern for Ulmo seedlings in this study. While both species are mid-tolerant to shade, the fact that one is deciduous and the other evergreen may have implications upon their responses to resource availability. Evergreen seedlings (such as Ulmo) are more conservative in terms of resource use (e.g., nutrients and water) than deciduous ones [59–63]. This can partly explain the lower sensitivity of Ulmo to the different restoration treatments and to increasing canopy openness.

### 4.3. Implications for Forests Restoration

Underplanting activities have been proven to be effective methods of artificial regeneration in many forested habitats around the globe [7], and therefore, it is a useful restoration practice to direct succession trajectories in a more predictable way [6,10,64]. The fast-growth patterns of Raulí and Ulmo would enable a more rapid forest recovery of high-graded forests, e.g., biomass accumulation in shorter periods than through natural succession [17]. Restoring the essential forest attributes is

pivotal to enhance the forest functions and processes and, not less important, the forest resilience when succession is arrested by recalcitrant understory vegetation after partial overstory disturbances [31], which is the case of the present study. Intermediate overstory densities have been proven to provide the best results in survival and growth of underplanted seedlings in most forest biomes in the world, including mid-tolerant species [7]. In this study, we did not evaluate the growth under very open canopies (>60%). However, Donoso et al. [65] and Soto et al. [10,35] showed that greater growth rates in Raulí occur in open fields when planted with neighboring faster-growing evergreen species (i.e., facilitation mechanisms under partial shade). In addition, Donoso et al. [9,45] showed that the growth in Raulí was indifferent to canopy openness during the seedling stages but during the sapling stages, growth was enhanced with increased light. Therefore, while further studies should continue to illustrate responses of underplanted seedlings to varying site and microsite conditions, this artificial regeneration option seems suitable for these species and convenient to restore high-graded forests.

## 5. Conclusions

Two mid-tolerant tree species that dominate the canopy in the forests where they naturally develop showed variable performances when underplanted in high-graded old-growth forests. Their mid-tolerant-to-shade character was reflected in the positive responses to increasing canopy openness, but Ulmo seemed to adapt better to these understory conditions, as reflected by a narrower variance of the growth within different levels of canopy openness. Raulí reacted positively to the restoration treatments and canopy openness, indicating the need of silviculture protocols in these high-graded forests through understory control and tree density reduction. Ecologically, the early results of this study may suggest that Ulmo has symmetric competition for light and Raulí is more sensitive to competition for soil nutrients and water (i.e., a positive effect of restoration treatments and canopy openness), suggesting asymmetric competition. The latter can be attributed to the evergreen nature of Ulmo in contrast to the deciduous Raulí.

**Author Contributions:** Conceptualization, D.P.S. and P.J.D.; methodology, P.J.D., D.P.S., and A.V.-G.; formal analysis, D.P.S., M.G.-C., and C.S.-E.; investigation, D.P.S., P.J.D., M.G.-C., and C.S.-E.; resources, P.J.D.; data curation, A.G.-G. and D.P.S.; writing—original draft preparation, D.P.S., P.J.D., and M.G.-C.; writing—review and editing, D.P.S., P.J.D., M.G.-C., C.S.-E., and A.V.-G.; visualization, D.P.S., M.G.-C., and C.S.-E.; supervision, P.J.D. and A.V.-G.; project administration, A.V.-G and P.J.D.; funding acquisition, P.J.D. All authors have read and agreed to the published version of the manuscript.

**Funding:** Fondo de Investigación del Bosque Nativo (FIBN) administered by the Corporación Nacional Forestal of Chile, grant number 005/2014.

**Acknowledgments:** D.P.S. and M.G.-C. thank the "Fondo Semilla 2018 de iniciación en la investigación de la Universidad de Aysén". We also thank students and research assistants that were involved in this research and, the constructive suggestions of 3 reviewers and the academic editor.

**Conflicts of Interest:** The authors declare no conflict of interest.

## Appendix A

### *Effects of the Structure Within the Random Effects*

Using the restricted estimated maximum likelihood (REML) after a stepwise selection approach, the significance of each component in the random terms were assessed. In general, for both species the proposed nested structure (i.e., the plot "P" within a site "site") for the random effect had a better performance than "site" and "P" alone (Table A1). Nevertheless, the slenderness models for Ulmo and Raulí, and the height growth model for Raulí, showed that the random structure with "site" performed better than the proposed nested ones (lowest AIC and BIC statistics). However, no significant differences among these models compared to the nested structure ($p > 0.05$) or having >2 AIC units have been found (Table A1).

**Table A1.** Restricted estimated maximum likelihood (REML) after a stepwise selection. Comparison between the best-supported models for each response variable (in bold) with those of the same structure in the fixed terms, with variation in the structure of the random terms. The (site/P) notation illustrates the nestedness of plots (P) within a site (Site).

| Species | Response Variable | Random Variable | AIC | BIC | $\chi^2$ | *p*-value |
|---|---|---|---|---|---|---|
| Ulmo | $pai_d$ | Site/P | −1313.7 | −1293.0 | - | - |
| Ulmo | $pai_d$ | Site | −1302.4 | −1285.9 | 13.2 | <0.001 |
| Ulmo | $pai_d$ | P | −1300.7 | −1284.2 | 14.9 | <0.001 |
| Ulmo | $pai_h$ | Site/P | 3626.6 | 3647.3 | - | - |
| Ulmo | $pai_h$ | Site | 3624.6 | 3641.1 | 52.636 | <0.001 |
| Ulmo | $pai_h$ | P | 3677.3 | 3693.8 | 0 | 1 |
| Ulmo | $pai_v$ | Site/P | 2273.4 | 2306.3 | - | - |
| Ulmo | $pai_v$ | Site | 2271.4 | 2300.2 | 0 | 1 |
| Ulmo | $pai_v$ | P | 2306.9 | 2335.7 | 35.503 | <0.001 |
| Ulmo | *Slen* | Site/P | 4086.6 | 4115.0 | - | - |
| Ulmo | *Slen* | Site | 4086.1 | 4119.3 | 1.5307 | 0.216 |
| Ulmo | *Slen* | P | 4200.3 | 4229.1 | 115.6 | <0.001 |
| Rauli | $pai_d$ | Site/P | −706.8 | −675.5 | - | - |
| Rauli | $pai_d$ | Site | −697.3 | −669.9 | 11.4 | <0.001 |
| Rauli | $pai_d$ | P | −678.4 | −650.9 | 30.4 | <0.001 |
| Rauli | $pai_h$ | Site/P | 3136.2 | 3179.3 | - | - |
| Rauli | $pai_h$ | Site | 3134.2 | 3173.4 | 0 | 1 |
| Rauli | $pai_h$ | P | 3153.5 | 3192.7 | 19.313 | <0.001 |
| Rauli | $pai_v$ | Site/P | 2153.4 | 2184.7 | - | - |
| Rauli | $pai_v$ | Site | 2174.3 | 2201.7 | 22.937 | <0.001 |
| Rauli | $pai_v$ | P | 2216.3 | 2243.7 | 64.916 | <0.001 |
| Rauli | *Slen* | Site/P | 3469.3 | 3500.6 | - | - |
| Rauli | *Slen* | Site | 3468.3 | 3495.7 | 0.9499 | 0.329 |
| Rauli | *Slen* | P | 3470.5 | 3497.9 | 3.19 | 0.074 |

Note: $pai_d$: periodic annual increment in root-collar diameter; $pai_h$: periodic annual increment in total height; $pai_v$: periodic annual increment in stem volume; *Slen*: slenderness index.

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
