# Peer review of "Differential Early Performance of Two Underplanted Hardwood Tree Species Following Restoration Treatments in High-Graded Temperate Rainforests"

_forests, doi:10.3390/f11040401_

Round 1

Reviewer 1 Report

Keywords

The use of ‘canopy openness’ and ‘GLMM’ in the keywords seems to be redundant. It would be better to include species names as they did not appear in the title.

Introduction

More information presented in the introduction on ecological requirements or even national genetic conservation programs of the studied species would improve the paper. The authors could provide precise information on the indexes of species light requirements as it is important for the interpretation of the results: Eucryphia cordifolia =2 (Lusk et al. 2011), Nothofagus alpine =3.

Results

I could not find Table 5. Some other references to the tables are also probably not correct. The use of stepwise selection approach should be presented in M&M.

Author Response

Response to reviewer #1

The use of ‘canopy openness’ and ‘GLMM’ in the keywords seems to be redundant. It would be better to include species names as they did not appear in the title.

Response: OK, we change as reviewer suggested.

Introduction

More information presented in the introduction on ecological requirements or even national genetic conservation programs of the studied species would improve the paper. The authors could provide precise information on the indexes of species light requirements as it is important for the interpretation of the results: Eucryphia cordifolia =2 (Lusk et al. 2011), Nothofagus alpine=3.

Response:

No one of the two tree species evaluated in this study are under any fragile conservation status, and we are not aware about any current genetic conservation program for either species. We have not significantly changed the introduction of the manuscript, but now we provide an entire new section in Discussion that deals with physiological traits of the two mid-tolerant species that determine their behavior in natural forests and may explain their differences in growth in the present study. This new section provides several new references.

Results

I could not find Table 5. Some other references to the tables are also probably not correct. The use of stepwise selection approach should be presented in M&M.

Response: Thank you, it was a mistake and now is solved.

Reviewer 2 Report

I couldn't jugde the originality and novelty of work because I'm not a specialist of South America silviculture.

The experiment seems to be well conducted and collected data elaborated in a correct way. There are some flaws in results presentation: Table 3 is difficult to read and must be formatted again.

Discussion of results should be improved because most of publication cited in this chapter are own publication of authors. Generally - there is a big share of self-citations in references.

Conclusions and recommendations for forest practice are well supported by results.

Author Response

Response to reviewer #2

I couldn't jugde the originality and novelty of work because I'm not a specialist of South America silviculture.

The experiment seems to be well conducted and collected data elaborated in a correct way. There are some flaws in results presentation: Table 3 is difficult to read and must be formatted again.

Response: We have formatted the Table 3 again. Thanks for your suggestion.  

Discussion of results should be improved because most of publication cited in this chapter are own publication of authors. Generally - there is a big share of self-citations in references.

Response: Yes, there are many citations of our own past work. We are sorry for this, but most of the works in underplanting of tree species within efforts of restoration or rehabilitation of high-graded forests conducted in Chile have been conducted by some of us.

Conclusions and recommendations for forest practice are well supported by results.

Reviewer 3 Report

Review of Soto et al. forests-745051

Differential early performance of two underplanted hardwood tree species following restoration treatments in high-graded temperate rainforests

This paper describes early results from a replicated forest restoration field experiment where underplanting of two important tree species was tested with and without additional silvicultural intervention. The paper is well written and the analysis appears sound and appropriate. Active restoration of degraded natural forests is an important topic, and this paper provides some early guidance on this for forest managers. I suggest only minor changes.

The Abstract and Results could be more useful and easy to interpret by managers simply by adding more quantitative results in the test (XX% improvement over controls” etc.) which help the reader look for important differences when they are referred to a figure or table. I’ve made some such suggestions in an annotated PDF accompanying this review (there are 55 separate annotations in this document).

Please describe the actual weed control treatment methods, and how much basal area or density was reduced by the improvement cutting. This will guide managers wanting to implement your treatments.

Please consider adding some more tangible results to compliment the figures, such as “Model coefficients indicated that seedlings receiving weed control (i.e., treatments C & D) were 7.4% taller and had 5% larger caliper than seedlings planted after improvement cutting (treatment B), and 12% taller and 15% larger caliper in the control treatments (A) (Figure 2). Better still would be to run post-hoc pairwise tests for significant differences among treatments (which treatments are different/same?). Also, consider pooling treatments C & D since scarification is presumably for a different study of natural regen, and not relevant for underplanting.

I propose evaluation of a fifth model: the null model, to see whether neither CO nor Treatment are significant in terms of lower AIC and BIC. This could be reported on in the text, or added to the tables, depending on the outcome of this test. If the null model has the lowest AIC, would this not be good outcome: i.e. ,suggests that managers have flexibility to underplant now and then wait for two years or more without need to implement other restoration activities?

Author Response

Response to reviewer #3

Differential early performance of two underplanted hardwood tree species following restoration treatments in high-graded temperate rainforests

This paper describes early results from a replicated forest restoration field experiment where underplanting of two important tree species was tested with and without additional silvicultural intervention. The paper is well written and the analysis appears sound and appropriate. Active restoration of degraded natural forests is an important topic, and this paper provides some early guidance on this for forest managers. I suggest only minor changes.

The Abstract and Results could be more useful and easy to interpret by managers simply by adding more quantitative results in the test (XX% improvement over controls” etc.) which help the reader look for important differences when they are referred to a figure or table. I’ve made some such suggestions in an annotated PDF accompanying this review (there are 55 separate annotations in this document).

Response: Thank you for your comment. We edited the results section in a deeper manner showing the % of improvements over restoration treatment following reviewer suggestions. However, we did not edit the abstract because we consider that more details would enlarge the abstract over 350 words, and we consider that potential readers can go through the paper if they are interested in more details, as reviewer pointed out. The changed text is highlighted with track of changes.

Please describe the actual weed control treatment methods, and how much basal area or density was reduced by the improvement cutting. This will guide managers wanting to implement your treatments.

Response: Thank you, we edited the text and table 2 accordingly. We highlighted it in the text with track of changes.  

Please consider adding some more tangible results to compliment the figures, such as “Model coefficients indicated that seedlings receiving weed control (i.e., treatments C & D) were 7.4% taller and had 5% larger caliper than seedlings planted after improvement cutting (treatment B), and 12% taller and 15% larger caliper in the control treatments (A) (Figure 2). Better still would be to run post-hoc pairwise tests for significant differences among treatments (which treatments are different/same?). Also, consider pooling treatments C & D since scarification is presumably for a different study of natural regen, and not relevant for underplanting.

Response: Thank you for this constructive comment. We edited the text as reviewer suggested. However, running a post hoc comparison with GLMM seems not possible until now, there are some R packages that can run it, but there are still a scientific discussion about it. Until now one can run post hoc test in mixed ANOVA and GLM but not GLMM, just is the case of our study.

I propose evaluation of a fifth model: the null model, to see whether neither CO nor Treatment are significant in terms of lower AIC and BIC. This could be reported on in the text, or added to the tables, depending on the outcome of this test. If the null model has the lowest AIC, would this not be good outcome: i.e. ,suggests that managers have flexibility to underplant now and then wait for two years or more without need to implement other restoration activities?

Response: interesting suggestion. We think that running a null model has not meaning for restoration practice when you have a control treatment, which is often used in restoration to compare treatments and control in well replicated experiment.